

# Evolution of tooth morphological complexity and its association with the position of tooth eruption in the jaw in non-mammalian synapsids

Tomohiro Harano and Masakazu Asahara

Division of Liberal Arts and Sciences, Aichi Gakuin University, Nisshin, Aichi, Japan

## ABSTRACT

Heterodonty and complex molar morphology are important characteristics of mammals acquired during the evolution of early mammals from non-mammalian synapsids. Some non-mammalian synapsids had only simple, unicuspid teeth, whereas others had complex, multicuspid teeth. In this study, we reconstructed the ancestral states of tooth morphological complexity across non-mammalian synapsids to show that morphologically complex teeth evolved independently multiple times within Therapsida and that secondary simplification of tooth morphology occurred in some non-mammalian Cynodontia. In some mammals, secondary evolution of simpler teeth from complex molars has been previously reported to correlate with an anterior shift of tooth eruption position in the jaw, as evaluated by the dentition position relative to the ends of component bones used as reference points in the upper jaw. Our phylogenetic comparative analyses showed a significant correlation between an increase in tooth complexity and a posterior shift in the dentition position relative to only one of the three specific ends of component bones that we used as reference points in the upper jaw of non-mammalian synapsids. The ends of component bones depend on the shape and relative area of each bone, which appear to vary considerably among the synapsid taxa. Quantification of the dentition position along the anteroposterior axis in the overall cranium showed suggestive evidence of a correlation between an increase in tooth complexity and a posterior shift in the dentition position among non-mammalian synapsids. This correlation supports the hypothesis that a posterior shift of tooth eruption position relative to the morphogenetic fields that determine tooth form have contributed to the evolution of morphologically complex teeth in non-mammalian synapsids, if the position in the cranium represents a certain point in the morphogenetic fields.

## INTRODUCTION

Heterodont dentition and complex molar morphology are evolutionary innovations that have enabled efficient food processing and adaptive radiations (*Romer & Parsons, 1986*; *Hunter & Jernvall, 1995*; *Colbert, Morales & Minkoff, 2001*; *Kemp, 2005*; *Luo, 2007*;

Corresponding author
Masakazu Asahara,
kamono.mana@gmail.com

*Ungar, 2010*) and are fundamental to mammalian diversity (*Weller, 1968*; *Stock et al., 1997*; *Feldhamer et al., 2003*; *Ungar, 2010*). These traits have been gradually acquired during the evolution of early mammals from some of the non-mammalian synapsids (*Weller, 1968*; *Feldhamer et al., 2003*; *Kemp, 2005*; *Ungar, 2010*). The acquisition process of complex molar morphology has been assumed to have progressed in the following sequence: one cusp, three main cusps, a triangular arrangement of main cusps, and then the addition of cusps at the lingual side in upper teeth and the buccal side in lower teeth, as indicated by the tritubercular theory (*Osborn, 1888*, *1897*; *Gregory, 1934*; *Crompton & Jenkins, 1968*; *Ungar, 2010*; *Yamanaka, 2022*). Therefore, the increase in the number of cusps and the change in their arrangement are considered major events in the evolution of mammalian-type molar morphology.

Synapsida includes Therapsida, which in turn includes Cynodontia (see Fig. 1 for the phylogeny of synapsid taxa considered in this study). Within Cynodontia, all mammals and their closest extinct relatives such as *Morganucodon* constitute Mammaliaformes; furthermore, Mammaliaformes and their closest extinct relatives including Tritylodontidae constitute Mammaliamorpha (*Rowe, 1988*). Early synapsids had only simple, unicuspid teeth, whereas non-mammalian cynodonts had complex, multicuspid teeth in heterodont dentitions (*Weller, 1968*; *Feldhamer et al., 2003*; *Kemp, 2005*; *Ungar, 2010*). The evolution of complex tooth morphology has been previously considered separately for some lower taxa of non-mammalian synapsids, with remarkable examples including the acquisition of leaf-shaped teeth as an adaptation to herbivorous diet in *Suminia* (*Rybczynski & Reisz, 2001*), the differentiation of a number of cusps on the teeth in *Dvinia* (*Tatarinov, 1968*; *Kemp, 1979*), and the occurrence of complex molar morphology in tritylodontids (*Kemp, 2005*). To our knowledge, no previous study has used recent phylogenetic hypotheses and comparative methods to reconstruct the evolutionary history of tooth morphology across non-mammalian synapsids.

Convergent evolution of several types of dental morphology is known to have occurred in mammals, with the most prominent examples including the independent acquisition of hypocones in several lineages with herbivorous adaptations (*Hunter & Jernvall, 1995*) and the independent evolution of carnassial teeth in several lineages with carnivorous adaptations (*Van Valkenburgh, 2007*; *Ungar, 2010*). In addition, tribosphenic molar function was independently acquired in both Laurasian and Gondwanan lineages of mammals during the Mesozoic (*i.e.*, Australosphenida and Boreosphenida), as well as in Shuotheriidae, and some docodont mammaliaforms (*Luo, 2007*). Among eutherian mammals, a simple molar morphology with the linear cusp arrangement or a single cusp have evolved secondarily in some taxa, such as pinnipeds and toothed whales (*Rybczynski, Dawson & Tedford, 2009*; *Armfield et al., 2013*; *Davis, 2019*; *Harano & Asahara, 2022a*). In squamate reptiles, a phylogenetic comparative study including fossil and extant taxa showed independent evolution of multicuspid teeth from a unicuspid ancestral morphology in a large number of lineages and reversals toward reduced tooth complexity in numerous lineages (*Lafuma et al., 2021*). Convergent evolution of a complex tooth morphology and secondary evolution of a simpler tooth morphology may have occurred in non-mammalian synapsids.

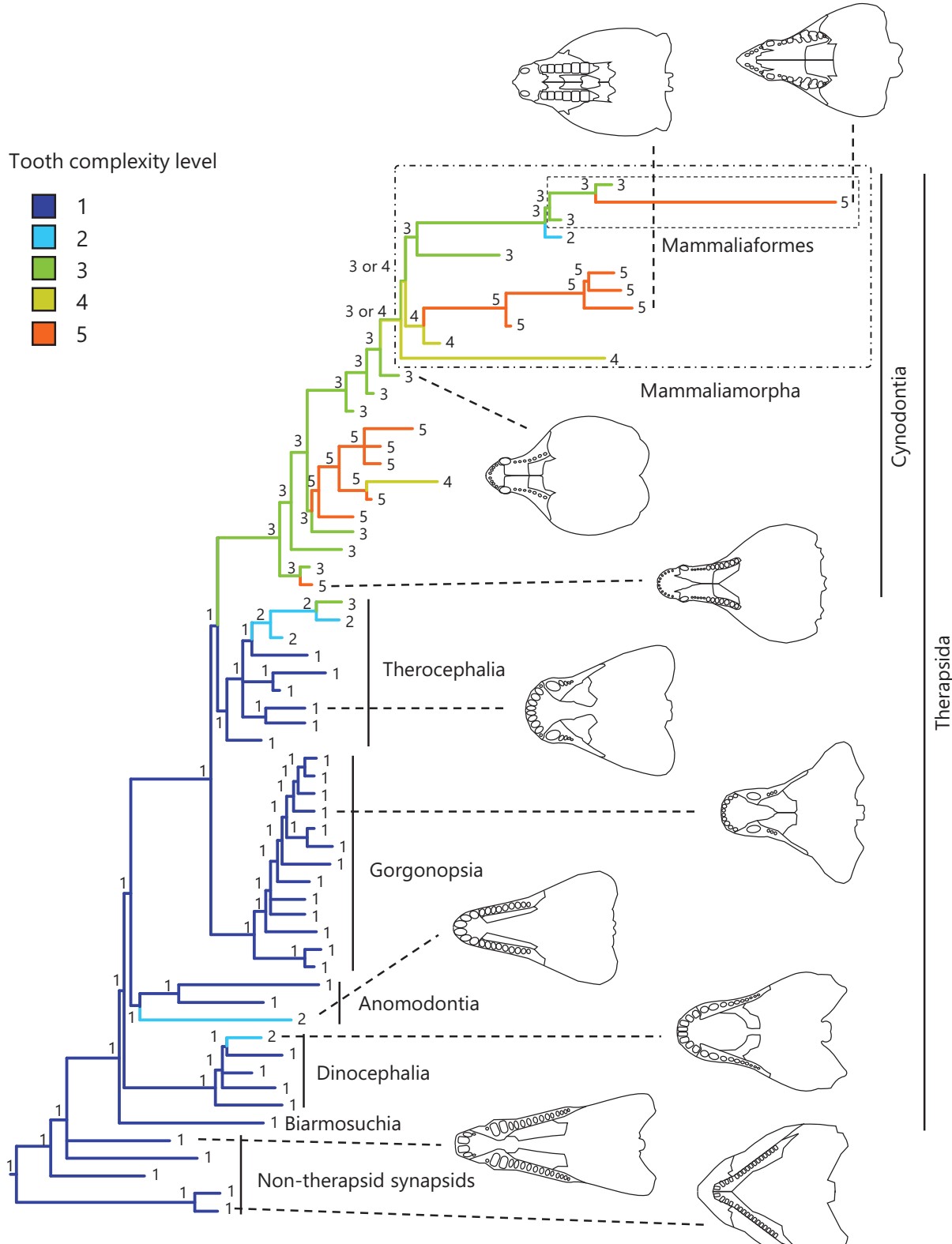

**Figure 1 Evolutionary history of tooth complexity on the phylogenetic tree of non-mammalian synapsids.** The ancestral states reconstructed using parsimony methods are indicated by the colors of the branches and the values assigned to the nodes. Schematic diagrams of ventral views of the cranium of the representative taxa show the approximate outline of the dentition, maxilla, and palatine. These diagrams were drawn based on

**Figure 1 (continued)**
illustrations taken from the literature: *Euromycter rutenus* (non-therapsid synapsids) from *Sigogneau-Russell & Russell (1974)*; *Dimetrodon limbatus* (non-therapsid synapsids) from *Amson & Laurin (2011)*; *Ulemosaurus svijagensis* (Dinocephalia) from *Kemp (2005)*; *Suminia getmanovi* (Anomodontia) from *Kemp (2005)*; *Leontosaurus vanderhorsti* (Gorgonopsia) from *Kammerer (2016)*; *Moschorhinus kitchingi* (Therocephalia) from *Durand (1991)*; *Dvinia prima* (Cynodontia) from *Tatarinov (1968)*; *Probainognathus jenseni* (Cynodontia) from *Hopson & Kitching (2001)*; *Kayentatherium* (Mammaliamorpha) from *Kemp (2005)*; and *Haldanodon exspectatus* (Mammaliaformes) from *Kielan-Jaworowska, Cifelli & Luo (2004)*. Additional details on the trees and the values of the ancestral states at each node are presented in Fig. S1.

The evolutionary and developmental factors underlying tooth morphological complexity have long been a focus of attention but are relatively poorly understood (*Cope, 1883*; *Osborn, 1888*, *1897*; *Tims, 1903*; *Butler, 1939*; *Feldhamer et al., 2003*; *Kemp, 2005*; *Ungar, 2010*; *Yamanaka, 2022*). Traditional field theory (*Butler, 1939*) postulates that morphogenetic fields in the jawbone induce different tooth forms, *i.e.*, the incisivization or caninization field in the anterior part of the jaw induces simple tooth morphology, whereas the molarization field in the posterior part of the jaw induces complex tooth morphology. Its modern version, the concentration gradients of morphogens, such as bone morphogenetic protein, BMP4, and fibroblast growth factor, FGF8, observed along the anteroposterior axis in the jawbone have been reported to be associated with tooth form. In the pig (*Sus scrofa*), BMP4 is expressed in the mesial (rostral) region, in which incisors or canines with simple morphology erupt, and FGF8 is expressed in the distal (caudal) region, in which morphologically complex molars erupt (*Armfield et al., 2013*). In contrast, BMP4 is expressed throughout the region of tooth eruption in the pantropical spotted dolphin (*Stenella attenuata*), which possesses only simple teeth (*Armfield et al., 2013*). In the house shrew (*Suncus murinus*), the BMP4 and FGF8 expression regions correspond to the incisor- and molar-forming regions, respectively, in the jaw (*Yamanaka et al., 2015*). In the gray short-tailed opossum (*Monodelphis domestica*), the regions with differential expression of homeobox genes, Alx3, Msx1, and BarX1, likely correspond to the regions of different tooth classes, and the expressions of BMP4 and FGF8 along the mesio-distal axis in the jaw overlap with those of Msx1 and BarX1, respectively (*Wakamatsu et al., 2019*). These morphogens and associated molecules are known to affect the shape and size of teeth and the size and number of cusps (*Asahara et al., 2016*; *Couzens et al., 2016*; *Couzens, Sears & Rücklin, 2019*; *Zurowski et al., 2018*; *Selig, Khalid & Silcox, 2021*). According to these findings, the tooth eruption position relative to the morphogenetic fields along the anteroposterior axis in the jaw appear to have an important role in determining tooth morphology.

Recently, *Harano & Asahara (2022a)* evaluated the morphological complexity of teeth and the anteroposterior position of dentition relative to the component bones in the upper jaws of numerous species of eutherian mammalian, including the carnivorans, cetaceans, and even-toed ungulates, and found a phylogenetically adjusted correlation between tooth simplification and an anterior shift (anteriorization) of the dentition position in the carnivoran clade (Carnivora) and the cetacean and even-toed ungulate clade (Artiodactyla, sometimes called Cetartiodactyla). Taken together, this finding, the field theory

(*Butler, 1939*), and the experimental evidence for morphogen concentration gradients associated with different tooth types (*Armfield et al., 2013*; *Yamanaka et al., 2015*; *Wakamatsu et al., 2019*) suggest that an anterior shift of the tooth eruption position relative to the morphogenetic fields, which are assumed to be present at specific locations associated with the component bones in the jaw, is a factor in the evolutionary simplification in tooth morphology in some carnivorans (*e.g.*, pinnipeds) and cetaceans (*i.e.*, toothed whales) (*Harano & Asahara, 2022a*).

The evolutionary simplification in tooth morphology in some eutherian mammals appears to have reversed the acquisition process of complex teeth that occurred during the evolution of early mammals from some of the non-mammalian synapsids (*Harano & Asahara, 2022a*). The evolution of complex teeth, as well as of simplified teeth, may be attributable to the shifting of the dentition position relative to the morphogenetic fields. Therefore, we hypothesized that a posterior shift of the dentition position relative to the morphogenetic fields in the jaw has contributed to the evolution of complex tooth morphology in non-mammalian synapsids. According to this hypothesis, tooth complexity can be expected to be correlated with the anteroposterior position of the dentition in synapsid jaws. To test this expectation, a comprehensive phylogeny of synapsids (*Jones, Angielczyk & Pierce, 2019*) was used in this study, along with phylogenetic comparative analyses that were conducted across a number of non-mammalian synapsids.

In this study, we rated tooth morphological complexity in non-mammalian synapsids and quantified the anteroposterior position of their dentition relative to the ends of component bones as reference points in the upper jaw, following a previous study (*Harano & Asahara, 2022a*). This approach assumed that the ends of component bones are a proxy for the positions of the morphogenetic fields. However, this assumption may not be valid for non-mammalian synapsids because the ends of component bones depend on the shape and relative area of each component bone in the upper jaw, and these appear to vary more remarkably among the synapsids (schematic diagrams of the cranium of the representative taxa are presented in Fig. 1) than among the carnivoran clade or the cetacean and even-toed ungulate clade in eutherian mammals. Therefore, the position of the dentition along the anteroposterior axis in the overall cranium was also quantified in non-mammalian synapsids. These measurements were used to reconstruct the evolutionary history of tooth morphological complexity and to test the correlation between tooth complexity and the anteroposterior position of the dentition in the upper jaw or the cranium.

## MATERIALS AND METHODS

### Measurements of dentition position

To measure the positioning of the dentition in the upper jaw or cranium, illustrations of ventral views of crania that were taken from the literature listed in Dataset S1 were used. These illustrations came from 60 taxa of non-mammalian synapsids, 55 of which belong to Therapsida. The Therapsida taxa included Biarmosuchia, Dinocephalia, Anomodontia, Gorgonopsia, Therocephalia, and Cynodontia. Of the 24 taxa of Cynodontia, 11 belong to Mammaliamorpha, and three belong to Mammaliaformes.

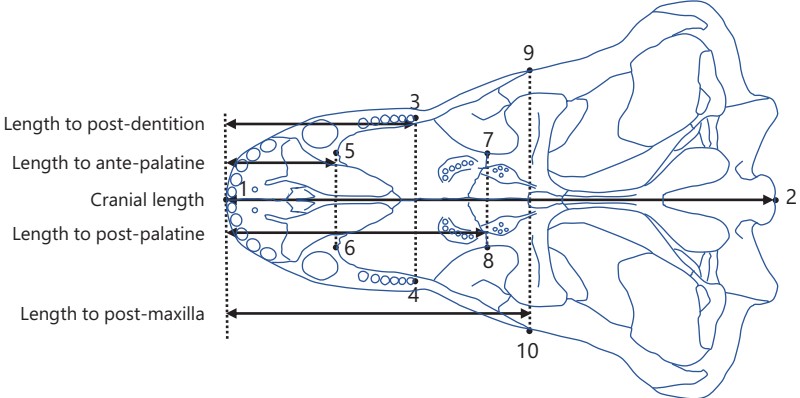

**Figure 2 Positions of landmarks used in this study.** The landmarks were defined as follows: (1) anterior end of the premaxilla; (2) basion; (3) and (4) posterior end of the dentition; (5) and (6) anterior end of the palatine; (7) and (8) posterior end of the palatine; (9) and (10) posterior end of the maxilla. The intersection of the midline of the cranium (the line connecting landmark 1 to 2) with the perpendicular line (denoted by the dotted lines) from each landmark (3 to 10) to the midline was taken to determine the anteroposterior position of each landmark. The distances between the anteroposterior positions were calculated as the lengths from the anterior end of the premaxilla to the posterior end of the dentition (length to post-dentition), the anterior end of palatine (length to ante-palatine), posterior end of palatine (length to post-palatine), posterior end of maxilla (length to post-maxilla), and posterior end of the cranium (cranial length). The diagram of the cranium used in this representation was drawn from an illustration of *Aelurognathus tigriceps* in *Kammerer (2016)*.

The posterior end of the dentition was used as a point to measure the dentition position (Fig. 2). In the upper jaw of mammals, the morphologically complex molars are rooted in the maxilla (*Yamanaka et al., 2015*). The regions of maxilla and the palatine adjacent to the maxilla were examined to determine reference points in the upper jaw of non-mammalian synapsids. The anterior end of palatine, posterior end of palatine, and posterior end of maxilla were identifiable in illustrations of crania of various non-mammalian synapsids and were used as the reference points (Fig. 2). Accordingly, the dentition position was evaluated relative to each of the three reference points. Moreover, the dentition position within the overall cranium was evaluated.

Measurements were conducted as previously described in *Harano & Asahara (2022a)*. Specifically, landmarks were digitized onto the illustrations using tpsDig software version 2.31 (*Rohlf, 2017*), with the configuration displayed in Fig. 2. A perpendicular line was drawn from each landmark to the midline of the cranium (the line connecting landmarks 1 and 2; Fig. 2), and the point of intersection was used to determine the position of the landmark along the anteroposterior axis. The length from the anterior end of the premaxilla to each of the posterior end of the dentition (length to post-dentition), the anterior end of palatine (length to ante-palatine), posterior end of palatine (length to post-palatine), and posterior end of maxilla (length to post-maxilla) were all calculated from the anteroposterior positions of landmarks on the left and right sides and then averaged for each specimen. Data on the length to post-maxilla were not available for two taxa (*Yunnanodon* and *Pseudotherium argentinus*) because the position of the posterior end of maxilla could not be identified in available illustrations. The length from the

Tooth complexity level

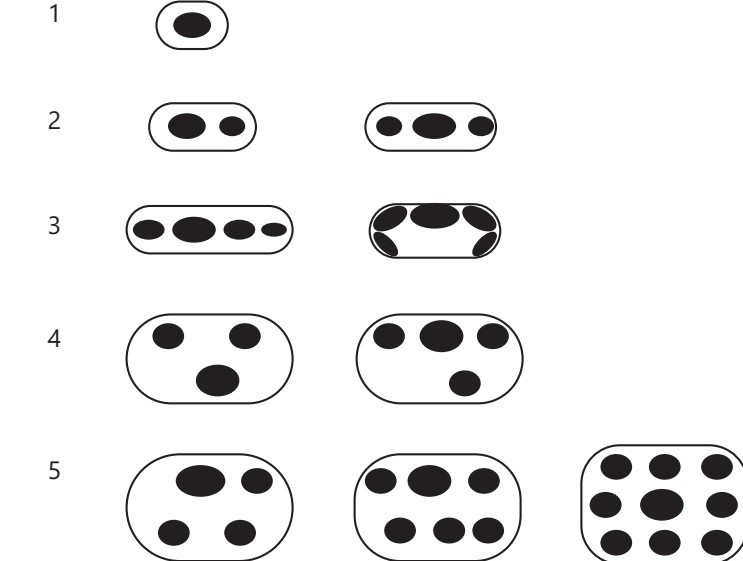

**Figure 3 Classification of tooth complexity used in this study.** Tooth complexity was categorized into five levels based on cusp number and arrangement. The schematic drawings represent occlusal views of the cusp arrangement on the tooth at each level of complexity.

anterior end of the premaxilla to the posterior end of the cranium (cranial length) were calculated for each specimen. If the scale was not provided with the illustration, available data on skull length of the taxon were used as a proxy for scale to allow the conversion of the distance seen in the illustration into an actual measurement usable for analyses. The measurements for all taxa included in the present study are shown in, Dataset S2. All data on length were converted to natural logarithms prior to subsequent analyses.

## Tooth complexity

Illustrations or images of teeth of the synapsid taxa included in this study were obtained from the literature listed in Dataset S1. The number and arrangement of cusps on the teeth shown in the illustration or image were visually examined to assess the tooth morphological complexity. Cusp number and arrangement have been used to classify tooth morphology in mammals (*Jernvall et al., 1996*; *Harano & Asahara, 2022a*) and mammaliaforms (*Couzens, Sears & Rücklin, 2021*), and cusp number has been used as a measure of tooth complexity in squamate reptiles (*Lafuma et al., 2021*). The criteria for classifying the tooth complexity level established in a previous study (*Harano & Asahara, 2022a*) were derived primarily from the stages originally proposed by the tritubercular theory (*Osborn, 1888*, *1897*; *Gregory, 1934*). We used these criteria with a few changes to accommodate any cusp arrangement observed in non-mammalian synapsids. Our criteria were as follows (Fig. 3):

Level 1: Only one cusp.

Level 2: Two or three cusps arranged in a row along the mesio-distal axis.

Level 3: More than three cusps arranged in a row along the mesio-distal axis or continuously in a row along the cingulum.

Level 4: Three cusps arranged in triangle form by shifting the cusps to the lingual or buccal side. If a tooth has more than two cusps, one of which was deviated from the row of cusps beyond the cusp width, then the tooth was classified in this level.

Level 5: Development of an additional cusp at the lingual side. If a tooth has the rows of cusps on both lingual and buccal side, then the tooth was classified in this level.

If the illustration or image showed only a lateral view of teeth, we considered that the cusps were arranged in a line along the mesio-distal axis and classified the teeth into either level 1, 2, or 3. Each taxon received a score of one of these five levels based on the highest level found in their teeth.

## Phylogenetic comparative analyses

The phylogeny of synapsids used in this study was derived from the supertree reconstructed by *Jones, Angielczyk & Pierce (2019)*, who described 60 time-scaled phylogenetic trees. A majority-rule consensus tree was computed with branch lengths from these 60 trees using the consensus edges function in the *phytools* package (*Revell, 2012*) of R version 4.2.2 (*R Development Core Team, 2022*). Taxa that were not included in our analyses were pruned from the tree. *P. argentinus* was inserted into the relevant positions in the tree according to *Wallace, Martínez & Rowe (2019)*. Among the taxa included in this study, *Hadrocodium wui* is the most closely related to mammals.

The ancestral states of tooth complexity level and the positioning of the dentition in the upper jaw or cranium were reconstructed using parsimony methods in Mesquite version 3.61 (*Maddison & Maddison, 2019*), as previously described in *Harano & Asahara (2022a)*. The tooth complexity level was treated as an ordered character. For an ordered character, parsimony methods determine the ancestral states that minimize the number of steps of character change, assuming that the number of steps from state i to state j is |i-j| (*Maddison & Maddison, 2019*). Thus, the underlying assumption for this approach is that all tooth complexity levels are placed at equal intervals. The positioning of the dentition was treated as a continuous character in the ancestral state reconstructions. To obtain univariate trait values of the positioning of the dentition, the length to post-dentition was regressed on each of the following: length to ante-palatine, length to post-palatine, length to post-maxilla, and cranial length (see Fig. 2 for length measurements); after which a residual value was calculated for each taxon. The residual values were used to quantify the dentition position relative to the anterior end of palatine (model formula: length to post-dentition (log) ~ length to ante-palatine (log)), relative to the posterior end of palatine (model formula: length to post-dentition (log) ~ length to post-palatine (log)), relative to the posterior end of maxilla (model formula: length to post-dentition (log) ~ length to post-maxilla (log)), and in the cranium (model formula: length to post-dentition (log) ~ cranial length (log)) because a larger residual value represents a more posterior positioning of the dentition. These regressions were conducted using phylogenetic

generalized least squares (PGLS) to control for the effect of phylogenetic relatedness between taxa. Pagel's lambda ($\lambda$), a measure of the phylogenetic signal, was estimated to scale the phylogenetic correlation structure in the PGLS models. In general, $\lambda$ ranges from 0 to 1, where 0 indicates no phylogenetic correlation and 1 indicates that the phylogenetic correlation structure is in agreement with a Brownian motion model of evolution. All PGLS analyses were performed using the phylolm function in the *phylolm* package (*Ho & Ané, 2014*) in R version 4.2.2 (*R Development Core Team, 2022*).

A correlation between tooth complexity and the positioning of the dentition in the upper jaw or cranium was tested following the methods previously described in *Harano & Asahara (2022a)*. Specifically, separate PGLS models were used to analyze the dentition position relative to the anterior end of palatine, relative to the posterior end of palatine, relative to the posterior end of maxilla, and in the cranium. The length to post-dentition was included as the response variable, and the level of tooth complexity and either the length to ante-palatine, length to post-palatine, length to post-maxilla, or cranial length were included as explanatory variables in the PGLS models (model formula: length to post-dentition (log) ~ tooth complexity level + length to ante-palatine (log), length to post-dentition (log) ~ tooth complexity level + length to post-palatine (log), length to post-dentition (log) ~ tooth complexity level + length to post-maxilla (log), and length to post-dentition (log) ~ tooth complexity level + cranial length (log); see Fig. 2 for definitions of positions and length measurements). These models enable the examination of the relationship between tooth complexity level and length to post-dentition while controlling for either the length to ante-palatine, length to post-palatine, length to post-maxilla, or cranial length as a predictor and phylogeny. To control for the effect of a predictor, including the predictor as one of the explanatory variables in a multiple regression model is more appropriate than taking the residual from the regression against the predictor and using the residual as data during further regression analysis (*García-Berthou, 2001*; *Freckleton, 2009*). The tooth complexity level was treated as a single quantitative variable in the PGLS models. This assumed that all tooth complexity levels are placed at equal intervals, similar to the ancestral state reconstructions using parsimony method described above.

## RESULTS

### Ancestral state reconstructions

To visualize the evolutionary history of tooth complexity, the reconstructed ancestral states were mapped onto the phylogenetic tree (Fig. 1; further details on the tree can be found in Fig. S1). All non-therapsid synapsids included in this analysis had the simplest, unicuspid teeth. The reconstruction showed that the most recent common ancestor of all therapsids inherited the simplest teeth from its ancestor. Within Therapsida, teeth with a second cusp (tooth complexity level 2) were independently acquired three times. Teeth with more than three cusps arranged in a line (tooth complexity level 3) evolved independently twice, one instance of which occurred at the base of Cynodontia. Within Cynodontia, the highest level of tooth complexity (tooth complexity level 5) evolved independently four times, two instances of which occurred in lineages that did not lead to Mammaliamorpha. The

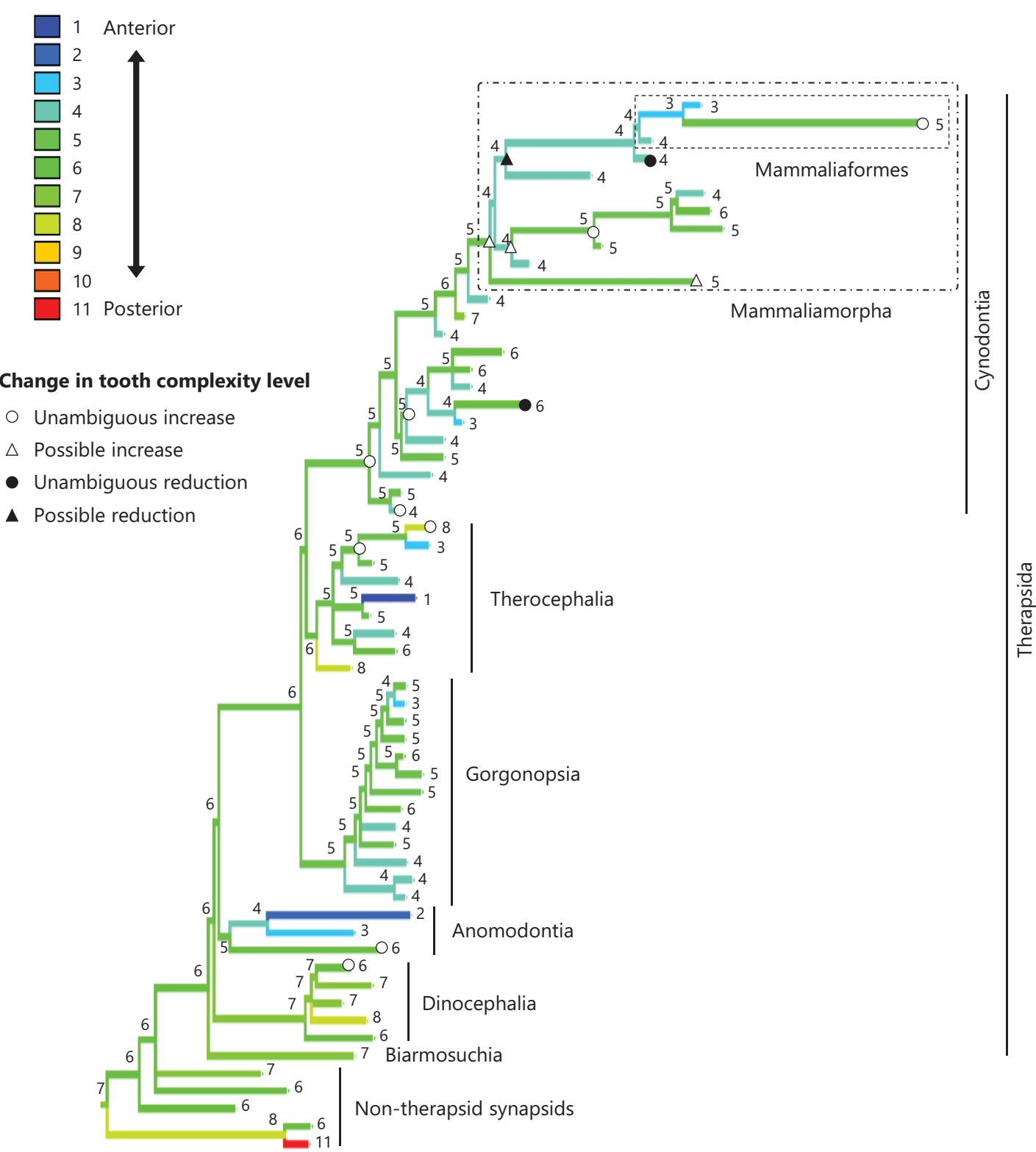

**Figure 4 Evolutionary history of the dentition position relative to the anterior end of palatine on the phylogenetic tree of non-mammalian synapsids.** This position was calculated as residuals from the PGLS regression of the length to post-dentition on the length to ante-palatine. The definitions of positions and length measurements are presented in Fig. 2. The statistics of the regression are presented in Table S1a. The ancestral

**Figure 4 (continued)**
states reconstructed using parsimony methods are indicated by the colors of the branches with numerical labels corresponding to each color. The open circles represent nodes of unambiguous increased tooth complexity, the open triangles represent nodes with increased tooth complexity in one of the most likely patterns for the evolutionary history of tooth complexity, the closed circle represents a node of unambiguous reduction in tooth complexity, and the closed triangle represents a node where reduction in tooth complexity was observed in one of the most likely patterns for the evolutionary history of tooth complexity. Additional details on the tree are presented in Fig. S2. The values of the ancestral states at each node can be found in Table S2.                                                                            

remaining two occurrences of independent evolution of the highest level of tooth complexity were found within Mammaliamorpha, one of which was found within Mammaliaformes. Within Mammaliamorpha, two possible patterns for the evolutionary history of teeth with triangular cusp arrangement (tooth complexity level 4) were equally supported. In one pattern, these teeth were independently acquired twice, while in the other, they were acquired at the base of Mammaliamorph and subsequently lost (evolution from tooth complexity level 4 to 3) at the base of the subclade including Mammaliaformes. Furthermore, reductions in tooth complexity occurred independently twice, with one instance (evolution from tooth complexity level 3 to 2) observed within Mammaliamorpha, and the other (evolution from tooth complexity level 5 to 4) occurring in another cynodont lineage.

The reconstructed ancestral states of the dentition position relative to the anterior end of palatine (Fig. 4; further details on the trees can be found in Fig. S2), relative to the posterior end of palatine (Fig. 5; further details on the trees can be found in Fig. S3), relative to the posterior end of maxilla (Fig. 6; further details on the trees can be found in Fig. S4), and in the cranium (Fig. 7; further details on the trees can be found in Fig. S5) were mapped onto the phylogenetic tree. The evolutionary history of these dentition positions appears to demonstrate considerable differences. The dentition position relative to the posterior end of maxilla (Fig. 6) and in the cranium (Fig. 7) showed a tendency to be more anterior in non-cynodont therapsids compared with the earlier, non-therapsid synapsids. These positions appear to have shifted posteriorly during the evolution of cynodonts from non-cynodont therapsids (Figs. 6 and 7). The dentition position in the cranium seems to have shifted further posteriorly in Mammaliamorpha (Fig. 7). Such shifts were not apparent in the evolutionary history of the dentition position relative to the anterior end of palatine (Fig. 4) and the posterior end of palatine (Fig. 5). An additional description of the reconstructed ancestral state of these positions is provided in Supplemental Text.

## Correlation between tooth complexity and the positioning of the dentition

No significant relationship was found between tooth complexity level and the length to post-dentition when controlled for the length to ante-palatine (Table 1A; Fig. S6A) and controlled for the length to post-palatine (Table 1B; Fig. S6B) across non-mammalian synapsids. A significant positive relationship between tooth complexity level and the length to post-dentition was found when controlled for the length to post-maxilla (Table 1C; Fig. S6C). A significant positive relationship was also observed between tooth complexity

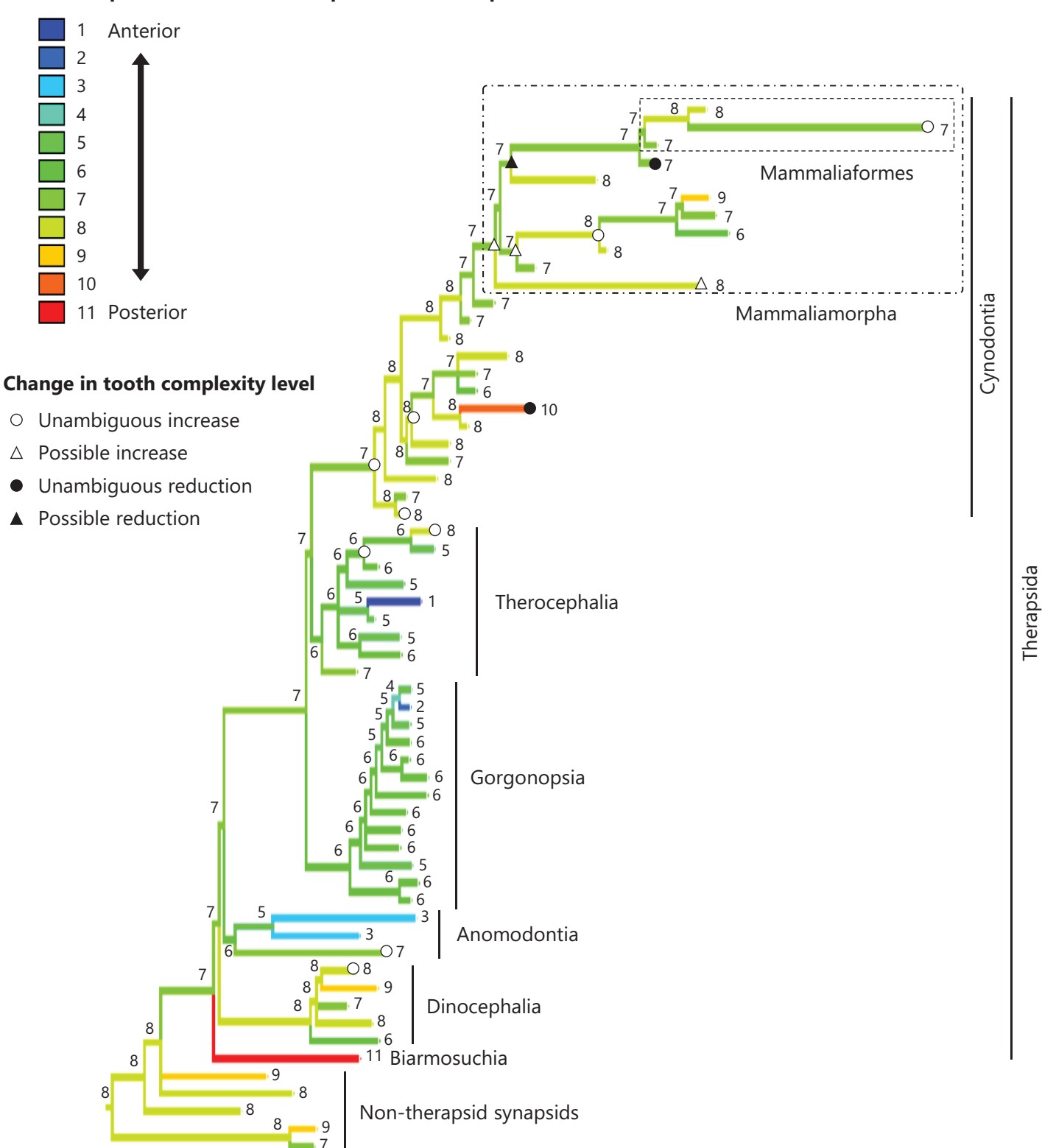

**Figure 5 Evolutionary history of the dentition position relative to the posterior end of palatine on the phylogenetic tree of non-mammalian synapsids.** This position was calculated as residuals from the PGLS regression of the length to post-dentition on the length to post-palatine. The definitions of positions and length measurements are presented in Fig. 2. The statistics of the regression are given in Table S1b. The ancestral

**Figure 5 (continued)**
states reconstructed using parsimony methods are indicated by the colors of the branches with numerical labels corresponding to each color. The open circles, open triangles, closed circle, and closed triangle are as in Fig. 4. Additional details on the tree can be found in Fig. S3. The values of the ancestral states at each node can be found in Table S3.               

level and the length to post-dentition when controlled for the cranial length (Table 1D; Fig. S6D). Therefore, an increase in tooth complexity level was significantly correlated with a posterior shift of the dentition position relative to post-maxilla, as well as with a posterior shift of the dentition position along the anteroposterior axis in the cranium across non-mammalian synapsids.

## DISCUSSION

This study reconstructed the evolutionary history of tooth morphological complexity on the phylogeny of non-mammalian synapsids to reveal that the common ancestor of all therapsids had the simplest, unicuspid teeth (tooth complexity of level 1), and complex teeth, characterized by the development of additional cusps and a triangular arrangement (tooth complexity of level 5), evolved independently in four lineages (Fig. 1). Furthermore, secondary simplification was estimated to have occurred independently in at least two lineages of non-mammalian cynodonts (Fig. 1). These results were obtained through the analyses performed in this study, which included a limited number of taxa. If a larger number of taxa are included in the analyses, it is possible that complex and secondarily simplified teeth could be estimated to have evolved more frequently in non-mammalian synapsids.

Tooth morphology is a relatively stable trait and is widely used in the phylogenetic estimation and classification of mammals (*Kangas et al., 2004*; *Ungar, 2010*; *Harjunmaa et al., 2014*). However, the number of cusps on the postcanine teeth varies among individuals in the ringed seal (*Phoca hispida*) (*Jernvall, 2000*). The outcomes of the present study indicate that the evolutionary changes in tooth morphological complexity have occurred independently multiple times in non-mammalian synapsids, in a similar manner as was inferred in mammals (*e.g.*, *Hunter & Jernvall, 1995*; *Couzens, Sears & Rücklin, 2021*; *Harano & Asahara, 2022a*) and squamate reptiles (*Lafuma et al., 2021*). These findings suggest that cusp differentiation may be influenced by changes in a small number of parameters involved in the developmental process of tooth morphogenesis (*Salazar-Ciudad & Jernvall, 2010*; *Asahara et al., 2016*; *Zurowski et al., 2018*; *Couzens et al., 2016*; *Couzens, Sears & Rücklin, 2019*; *Selig, Khalid & Silcox, 2021*).

Simple tooth variants appear more readily than complex tooth variants, which generates a bias against increase in tooth complexity (*Harjunmaa et al., 2012*). Nevertheless, an evolutionary increase rather than a decrease in tooth complexity was predominant in mammals (*Hunter & Jernvall, 1995*; *Ungar, 2010*; *Couzens et al., 2016*; *Couzens, Sears & Rücklin, 2021*). A similar pattern was observed in non-mammalian synapsids. This pattern is not solely due to evolution beginning from the lowest tooth complexity (tooth complexity of level 1), where only increases in tooth complexity can occur. Our study showed that in teeth of intermediate complexity (tooth complexity of level 2, 3, or 4),

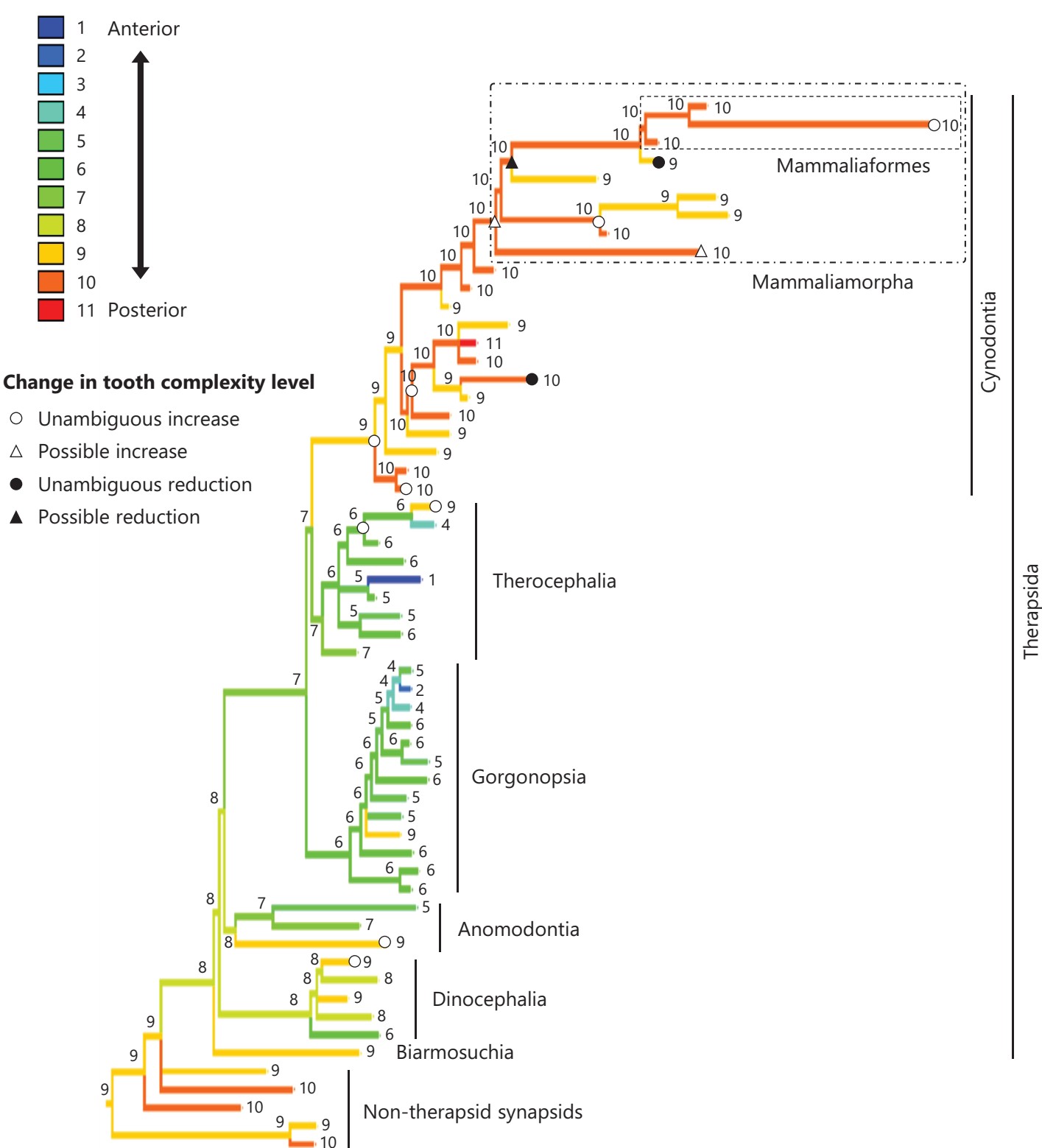

**Figure 6  Evolutionary history of the dentition position relative to the posterior end of maxilla on the phylogenetic tree of non-mammalian synapsids.** This position was calculated as residuals from the PGLS regression of the length to post-dentition on the length to post-maxilla.

**Figure 6 (continued)**
The definitions of positions and length measurements are presented in Fig. 2. The statistics of the regression are given in Table S1c. The ancestral states reconstructed using parsimony methods are indicated by the colors of the branches with numerical labels corresponding to each color. The open circles, open triangles, closed circle, and closed triangle are as in Fig. 4. Additional details on the tree can be found in Fig. S4. The values of the ancestral states at each node can be found in Table S4.

where both increases and decreases in tooth complexity are possible, an increase in tooth complexity occurred at least six times, while a decrease occurred once or twice (Fig. 1). The morphologically complex molars of mammals have shearing, crushing or grinding function that facilitates food processing through mastication (*Ungar, 2010*; *Williams, 2019*). Selection for improved ability to process foods appears to overcome the bias in the production of variation in tooth morphology, thereby resulting in more frequent evolution toward complex teeth across synapsids including mammals.

In eutherian mammals, the simplification of molar morphology is correlated with the anteriorization of the dentition position relative to component bones in the upper jaw in the carnivoran clade and the cetacean and even-toed ungulate clade (*Harano & Asahara, 2022a*). This implies that shifting the dentition position relative to the morphogenetic fields along the anteroposterior axis in the jaw, which are assumed to be present at specific locations associated with the component bones in the jaw, has contributed to the evolution of tooth complexity (*Harano & Asahara, 2022a*). Our phylogenetic comparative analyses failed to detect a significant correlation between tooth complexity and the dentition position relative to two of the three specific ends of component bones used as reference points in the upper jaw of non-mammalian synapsids (Tables 1A, 1B). This study included a limited number of taxa of non-mammalian synapsids, as data on the dentition position relative to the ends of component bones in the upper jaw were only available for these taxa. This limitation might have hindered the detection of a significant correlation between tooth complexity and the dentition position. Nevertheless, our analyses showed a significant correlation between more complex tooth morphology and a more posterior dentition position relative to the remaining one of the three specific ends of component bones in the upper jaw of non-mammalian synapsids (Table 1C). The reconstructed evolutionary history of the dentition positioning in the upper jaw varied considerably according to the reference point (Figs. 4–6). These results are attributable to the fact that the positional relationships between the ends of component bones vary among the taxa (Fig. S7).

The ends of component bones depend on the shape and relative area of each component bone in the upper jaw (see Fig. 1), whereas the position of the dentition along the anteroposterior axis in the overall cranium does not. A more complex tooth morphology was significantly correlated with a more posterior dentition position along the anteroposterior axis in the cranium across non-mammalian synapsids (Table 1D). The morphogenetic fields may be associated with the specific location along the anteroposterior axis in the cranium rather than specific bone locations in non-mammalian synapsids. If this is the case, our results are consistent with the expectation from the hypothesis that a

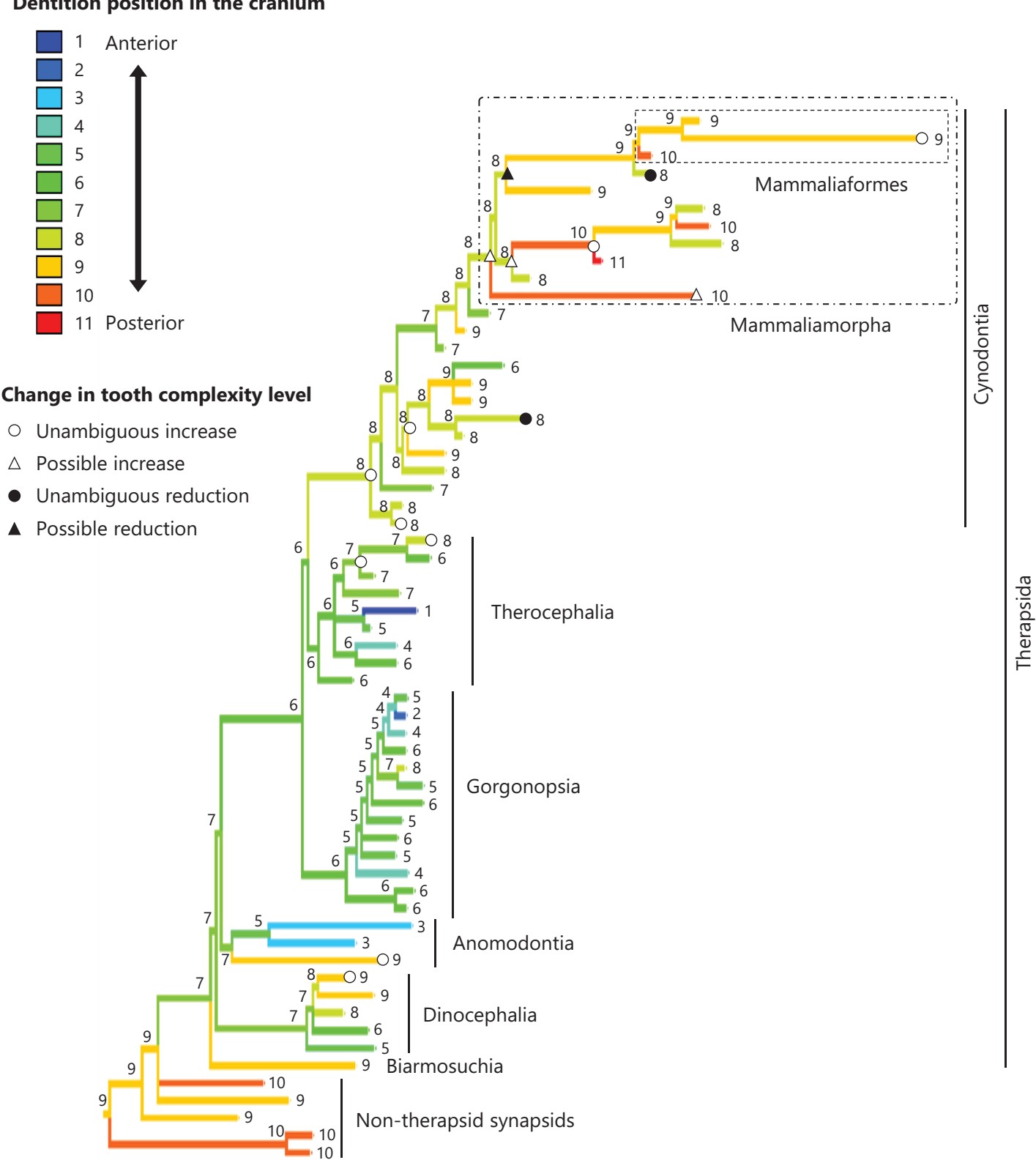

**Figure 7 Evolutionary history of the dentition position in the cranium on the phylogenetic tree of non-mammalian synapsids.** This position was calculated as residuals from the PGLS regression of the length to post-dentition on the cranial length. The definitions of positions and length

# PeerJ

**Figure 7 (continued)**
measurements are presented in Fig. 2. The statistics of the regression are given in Table S1d. The ancestral states reconstructed using parsimony methods are indicated by the colors of the branches with numerical labels corresponding to each color. The open circles, open triangles, closed circle, and closed triangle are as in Fig. 4. Additional details on the tree can be found in Fig. S5. The values of the ancestral states at each node can be found in Table S5.                         

**Table 1 Estimates of phylogenetic generalized least squares (PGLS) models investigating the relationship between tooth complexity and length to post-dentition when controlled for either the length to ante-palatine, length to post-palatine, length to post-maxilla, or cranial length across non-mammalian synapsids.**

|     | Explanatory variable | Estimate | SE | t | P |
|-----|----------------------|----------|-----|-----|-----|
| (a) | Intercept | 1.097 | 0.251 | 4.371 | <0.001 |
|     | Length to ante-palatine | 0.902 | 0.057 | 15.819 | <0.001 |
|     | Tooth complexity | −0.001 | 0.044 | −0.020 | 0.984 |
| (b) | Intercept | −0.053 | 0.210 | −0.252 | 0.802 |
|     | Length to post-palatine | 0.988 | 0.037 | 26.938 | <0.001 |
|     | Tooth complexity | 0.043 | 0.031 | 1.370 | 0.176 |
| (c) | Intercept | −0.146 | 0.169 | −0.865 | 0.391 |
|     | Length to post-maxilla | 0.990 | 0.031 | 31.572 | <0.001 |
|     | Tooth complexity | 0.068 | 0.025 | 2.699 | 0.009 |
| (d) | Intercept | −0.941 | 0.217 | −4.328 | <0.001 |
|     | Cranial length | 1.043 | 0.037 | 28.400 | <0.001 |
|     | Tooth complexity | 0.069 | 0.029 | 2.372 | 0.021 |

**Note:**
(a) The length to post-dentition (log) was included as the response variable, while the tooth complexity level and the length to ante-palatine (log) were included as explanatory variables (see Fig. 2 for definitions of positions and length measurements). An estimated $\lambda$ of 0.527 was used in the PGLS model. (b) The length to post-dentition (log) was included as response variable, while the tooth complexity level and the length to post-palatine (log) were included as explanatory variables. An estimated $\lambda$ of 0.919 was used in the PGLS model. (c) The length to post-dentition (log) was included as response variable, while the tooth complexity level and the length to post-maxilla (log) were included as explanatory variables. An estimated $\lambda$ of 0.777 was used in the PGLS model. (d) The length to post-dentition (log) was included as the response variable, while the tooth complexity level and the cranial length (log) were included as explanatory variables. An estimated $\lambda$ of 0.839 was used in the PGLS model.

posterior shift of the dentition position relative to the morphogenetic fields is a factor in the evolution of morphologically complex teeth.

The dentition position along the anteroposterior axis in the cranium can be affected not only by its position within the upper jaw but also by braincase size, and because the posterior expansion of braincase may move the dentition relatively forward in the cranium. The morphogenetic fields that were supposed originally in the field theory (*Butler, 1939*) could be interpreted to represent the concentration gradients of morphogens, such as BMP4 and FGF8 (*Armfield et al., 2013*). If future studies identify a metric that corresponds to the morphogen concentration gradient of the jaw, then it would be possible to more rigorously test a correlation between the tooth complexity and the dentition position relative to the morphogenetic fields in non-mammalian synapsids.

In the reconstructed evolutionary history, the dentition position showed a tendency to be more posterior relative to the posterior end of maxilla (Fig. 6) and in the cranium (Fig. 7) in non-therapsid synapsids compared with non-cynodont therapsids and early

cynodonts. In contrast, the ancestral simplest tooth morphology was retained throughout non-therapsid synapsids (Fig. 1). Therefore, the shifting of dentition position did not affect tooth morphology in non-therapsid synapsids. Among the synapsid taxa included in this study, teeth with multiple cusps (tooth complexity of level 2 or higher) were acquired within the therapsid clade and evolution toward more complex teeth (tooth complexity of level 3 or higher) occurred within the therocephalian and cynodont clades (Fig. 3). The dentition position in the cranium appears to have shifted posteriorly during the evolution from non-cynodont therapsids to Mammaliamorpha (Fig. 7). Changes in developmental mechanisms that allow the morphogen concentration gradient to affect tooth morphology may have occurred in early therapsids.

A posterior shift of the dentition position in the cranium brings the teeth closer to the jaw joint, which increases the mechanical advantage by converting the force of the masticatory muscle to a higher bite force (*Greaves, 1978*; *Thomason, 1991*). The dietary habit of consuming mechanically resistant foods has been suggested to favor the evolution of the molars closer to the jaw joint in carnivoran mammals (*Harano & Asahara, 2022b*). Selection for improved ability to process foods may have caused the evolution of complex teeth in correlation with a more posterior dentition position in non-mammalian synapsids. Furthermore, this selection may have facilitated the evolution of the novel jaw joint between the dentary and squamosal bones, which enables the precise occlusion between the upper and lower teeth and thereby efficient mastication (*Kemp, 2006*; *Tucker, 2017*; *Navarro-Díaz, Esteve-Altava & Rasskin-Gutman, 2019*).

## CONCLUSIONS

The present study revealed that morphologically complex teeth evolved independently multiple times and that the reversible evolution occurred in non-mammalian synapsids, as is also found in mammals (*Couzens, Sears & Rücklin, 2021*; *Harano & Asahara, 2022a*). Our analyses indicated suggestive evidence of a correlation between a more complex tooth morphology and a more posterior dentition position relative to one of the three specific bone locations used as reference points in the upper jaw across non-mammalian synapsids. Furthermore, quantification of the dentition position in the cranium revealed similar suggestive evidence. This finding provides conditional support for the hypothesis that a posterior shift of the dentition position relative to the morphogenetic fields in the jaw has contributed to the evolution of morphologically complex teeth in non-mammalian synapsids. Future studies will need to more closely examine the developmental mechanisms linking a posterior shift of the dentition and changes in tooth complexity during evolution.

## ACKNOWLEDGEMENTS

We thank Christian Kammerer and the anonymous reviewer for their valuable comments on the manuscript.

### Funding

This work was supported by JSPS KAKENHI Grant Number JP 19H03290 to Masakazu Asahara. The funders had no role in study design, data collection and analysis, decision to publish, or preparation of the manuscript.

### Grant Disclosures

The following grant information was disclosed by the authors:
JSPS KAKENHI: JP 19H03290.

### Competing Interests

The authors declare that they have no competing interests.

### Author Contributions

- Tomohiro Harano conceived and designed the experiments, performed the experiments, analyzed the data, prepared figures and/or tables, authored or reviewed drafts of the article, and approved the final draft.
- Masakazu Asahara conceived and designed the experiments, authored or reviewed drafts of the article, and approved the final draft.

### Data Availability

  The measurement data is available in the Supplemental File.

### Supplemental Information

Supplemental information for this article can be found online at http://dx.doi.org/10.7717/peerj.17784#supplemental-information.

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
