# Peer review of "Evolution of tooth morphological complexity and its association with the position of tooth eruption in the jaw in non-mammalian synapsids"

_PeerJ, doi:10.7717/peerj.17784_

## Round 0.1 · original submission · Minor Revisions

This manuscript has now received two reviews, including one from a previous reviewer (Reviewer 2 - presumably for another journal). While Reviewer 1 recommended major revisions, the previous reviewer found improvements and thought that the general conclusions mapped to the work. On the whole, I tend towards the minor revisions part of the spectrum but hold my final decision on publication dependent on how well the authors can address Reviewer 1's concerns (especially on sampling, immediately below).

The major crux of Reviewer 1's issues center on taxonomic sampling and its effect on appropriately characterizing the dental complexity of the therapsid clades under study. The authors need to address this issue in a satisfactory way in their revisions by either a) saying why they don't need to include greater sampling or b) actually improving their sampling, and updating their analyses. (I note that the authors do concede on this point -- see ll. 335 of the manuscript -- but it is clearly worthy of elaboration). While increasing sampling may take time, it should not materially change the conclusions (although it'd be interesting if it does!). By extension, the authors also need to weigh on the implication of their sampling: that "the lack of significant correlation between tooth complexity and dentition position is an artifact of limited sampling of complex-toothed clades." The authors need to respond in a satisfactory way to this point.

Lastly, a quick scan of the manuscript shows a small error (ll. 116, 148): Cetartiodactyla should not be mentioned as an order. In general, I encourage the authors to continue their focus on clade names and not the ranks (which are arbitrary). More specific rationale can be read in Prothero et al. (2021, J. Mamm. Evol.).

The revised manuscript should address all of the issues spotlighted by the reviewers and will be sent out for subsequent review.

·

Basic reporting

Paper is clearly-written and figures are generally well-constructed (although there seem to be some unusual processing artifacts on Figure 2; I am unsure if that is original to the figure or just a problem with the reviewing PDF).

Experimental design

This is an interesting approach to understanding the history of the mammalian dentition, and I think the authors have some intriguing ideas on the importance of posteriorization in synapsid dental evolution. However, this study is substantially hindered by poor sampling and mischaracterization of the dentition of several important taxa (based on inaccurate historic illustrations rather than direct observation of the fossils). The three major non-cynodont therapsid clades exhibiting complex crown morphologies have only a single sampled representative apiece (Suminia for non-dicynodont anomodonts, Ulemosaurus for tapinocephalid dinocephalians, and Bauria for bauriamorph therocephalians). Bauria is mistakenly classified as tooth complexity 1 despite having expanded multicusped postcanines with complex occlusion equal to that of many cynodonts. Bauriamorph therocephalians with even more complex dentition than Bauria (probably representing "level 5" complexity under the authors' classification), such as Hazhenia and Ordosiodon, are not even considered. Also notable in its paucity is sampling in Gomphodontia, the non-mammaliamorph cynodont clade with the most complex dentitions, which is a species-rich clade with a high diversity of cranial proportions. Would sampling more longirostrine traversodontid and trirachodontid (a clade completely unsampled in the current project) taxa yield different results regarding posteriorization? Why is Gorgonopsia, the therapsid clade with by far the most boring and homomorphic teeth, the most heavily sample non-cynodont group? (other than that I did a good job of illustrating many of the known taxa in palatal view). How do the authors account for forms of tooth complexity beyond simple cusp number, such as the existence of multiple tooth rows in certain anomodonts (e.g. Biseridens, Endothiodon), or tooth rows set on non-marginal elements such as the ectopterygoid tooth row of anomocephalids? I recognize that truly comprehensive sampling of a major clade is impossible, there is always more to sample, and one has to cut it off somewhere. It is also time-consuming and expensive to personally examine all the fossils in question. But in its present state I cannot help but feel that this data set is insufficient to address the question the authors are asking--even just within the published literature there are many more papers that illustrate well the dentition of synapsid taxa, which should have be included (see Hopson, 2005, 2014; Sidor & Hopson, 2017 for examples of gomphodont cynodonts with very well-illustrated dentitions, for example).

Validity of the findings

The data itself is well-analyzed and the authors' conclusions follow accordingly. I certainly agree with the conclusion that complex teeth evolved multiple times in synapsid evolution (this is already well known), which remains evident even with minimal sampling of non-cynodont complex-toothed clades. However, I wonder whether the lack of significant correlation between tooth complexity and dentition position is an artifact of limited sampling of complex-toothed clades. When you only sample one representative with complex teeth between morphologically extremely disparate groups, it is perhaps unsurprising that few commonalities emerge. But what about within bauriamorphs relative to other therocephalians, or tapinocephalids relative to other dinocephalians, etc.? Pelycosaurs seem like an afterthought as well, despite substantial variation in tooth position even within clades (compare Gordodon with Edaphosaurus). It really all comes back this.

Reviewer 2 ·

Basic reporting

L353: delete ‘so’.

Experimental design

Figs 4-7: please place a key to the symbols in the figure rather than relying on the caption (the information is only given in the caption for Fig. 4).

Validity of the findings

L351: some point should be made here that if you start from the minimum tooth complexity only increases in tooth complexity are possible (at least at the beginning) and so the trend for increasing tooth complexity is partly due to starting at the lower bound.
L389: it is more accurate to say that higher complexity of teeth is correlated with more posterior position without reference to the position of morphogenetic fields (which is only inferred here).
L395: the ‘could now’ seem to imply that this statement is a new interpretation, but has existed since before Armfield et al. (2013) and the current manuscript.

Additional comments

This version of the manuscript is an improvement on one I have previously reviewed. As long as it is clear that no definitive statement/conclusions about the relative position of morphogenetic fields, this is a worthwhile contribution to the study of the evolution of tooth complexity in vertebrates.

---

## Round 0.2 · accepted · Accept

The revised manuscript received a very favorable review from a previous reviewer, who found essentially no issues and recommended publication. Reading over the rebuttal and the revised manuscript, I agree with this assessment: this revised manuscript addresses all of the issues that were concerns, including taxonomic sampling and its bearing on their analyses. This manuscript is now ready for publication, and I thank the authors for their hard work, which has improved the content of this study through the review process.

Reviewer 2 ·

Basic reporting

Language of changes responding to my comments is clear and adequate.

Experimental design

No issues

Validity of the findings

No issues